# LATENT CONSTRAINTS:
# LEARNING TO GENERATE CONDITIONALLY FROM UNCONDITIONAL GENERATIVE MODELS

**Jesse Engel**
Google Brain
San Francisco, CA, USA

**Matthew D. Hoffman**
Google Inc.
San Francisco, CA, USA

**Adam Roberts**
Google Brain
San Francisco, CA, USA

## ABSTRACT

Deep generative neural networks have proven effective at both conditional and unconditional modeling of complex data distributions. Conditional generation enables interactive control, but creating new controls often requires expensive retraining. In this paper, we develop a method to condition generation without retraining the model. By post-hoc learning *latent constraints*, value functions that identify regions in latent space that generate outputs with desired attributes, we can conditionally sample from these regions with gradient-based optimization or amortized actor functions. Combining attribute constraints with a universal "realism" constraint, which enforces similarity to the data distribution, we generate realistic conditional images from an unconditional variational autoencoder. Further, using gradient-based optimization, we demonstrate identity-preserving transformations that make the minimal adjustment in latent space to modify the attributes of an image. Finally, with discrete sequences of musical notes, we demonstrate zero-shot conditional generation, learning latent constraints in the absence of labeled data or a differentiable reward function.

## 1 INTRODUCTION

Generative modeling of complicated data such as images and audio is a long-standing challenge in machine learning. While unconditional sampling is an interesting technical problem, it is arguably of limited practical interest in its own right: if one needs a non-specific image (or sound, song, document, etc.), one can simply pull something at random from the unfathomably vast media databases on the web. But that naive approach may not work for *conditional* sampling (i.e., generating data to match a set of user-specified attributes), since as more attributes are specified, it becomes exponentially less likely that a satisfactory example can be pulled from a database. One might also want to *modify* some attributes of an object while preserving its core identity. These are crucial tasks in creative applications, where the typical user desires fine-grained controls (Bernardo et al., 2017).

One can enforce user-specified constraints at training time, either by training on a curated subset of data or with conditioning variables. These approaches can be effective if there is enough labeled data available, but they require expensive model retraining for each new set of constraints and may not leverage commonalities between tasks. Deep latent-variable models, such as Generative Adversarial Networks (GANs; Goodfellow et al., 2014) and Variational Autoencoders (VAEs; Kingma & Welling, 2013; Rezende et al., 2014), learn to unconditionally generate realistic and varied outputs by sampling from a semantically structured latent space. One might hope to leverage that structure in creating new conditional controls for sampling and transformations (Brock et al., 2016).

Here, we show that new constraints can be enforced post-hoc on pre-trained unsupervised generative models. This approach removes the need to retrain the model for each new set of constraints, allowing users to more easily define custom behavior. We separate the problem into (1) creating an unsupervised model that learns how to reconstruct data from latent embeddings, and (2) leveraging the latent structure exposed in that embedding space as a source of prior knowledge, upon which we can impose behavioral constraints.

Our key contributions are as follows:

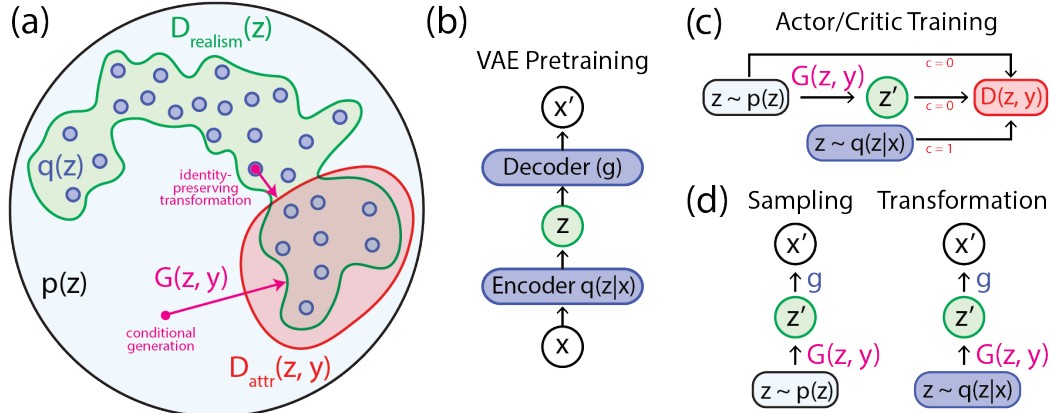

Figure 1: (a) Diagram of latent constraints for a VAE. We use one critic $D_{\text{attr}}$ to predict which regions of the latent space will generate outputs with desired attributes, and another critic $D_{\text{realism}}$ to predict which regions have high mass under the marginal posterior, $q(z)$, of the training data. (b) We begin by pretraining a standard VAE, with an emphasis on achieving good reconstructions. (c) To train the actor-critic pair we use constraint-satisfaction labels, $c$, to train $D$ to discriminate between encodings of actual data, $z \sim q(z|x)$, versus latent vectors $z \sim p(z)$ sampled from the prior or transformed prior samples $G(z \sim p(z), y)$. Similar to a Conditional GAN, both $G$ and $D$ operate on a concatenation of $z$ and a binary attribute vector, $y$, allowing $G$ to learn conditional mappings in latent space. If $G$ is an optimizer, a separate attribute discriminator, $D_{\text{attr}}$ is trained and the latent vector is optimized to reduce the cost of both $D_{\text{attr}}$ and $D_{\text{realism}}$. (d) To sample from the intersection of these regions, we use either gradient-based optimization or an amortized generator, $G$, to shift latent samples from either the prior ($z \sim p(z)$, sampling) or from the data ($z \sim q(z|x)$, transformation).

- We show that it is possible to generate conditionally from an unconditional model, learning a critic function $D(z)$ in latent space and generating high-value samples with either gradient-based optimization or an amortized actor function $G(z)$, even with a non-differentiable decoder (e.g., discrete sequences).

- Focusing on VAEs, we address the tradeoff between reconstruction quality and sample quality (without sacrificing diversity) by enforcing a universal "realism" constraint that requires samples in latent space to be indistinguishable from encoded data (rather than prior samples).

- Because we start from a VAE that can reconstruct inputs well, we are able to apply *identity-preserving transformations* by making the minimal adjustment in latent space needed to satisfy the desired constraints. For example, when we adjust a person's expression or hair, the result is still clearly identifiable as the same person (see Figure 5). This contrasts with pure GAN-based transformation approaches, which often fail to preserve identity.

- *Zero-shot conditional generation.* Using samples from the VAE to generate exemplars, we can learn an actor-critic pair that satisfies user-specified rule-based constraints in the absence of any labeled data.

## 2 BACKGROUND

Decoder-based deep generative models such as VAEs and GANs generate samples that approximate a population distribution $p^{\star}(x)$ by passing samples from some simple tractable distribution $p(z)$ (often $p(z) \triangleq \mathcal{N}(0, I)$) through a deep neural network. GANs are trained to fool an auxiliary classifier that tries to learn to distinguish between real and synthetic samples. VAEs are fit to data using a variational approximation to maximum-likelihood estimation:

$$\mathcal{L}^{\text{ELBO}} \triangleq \frac{1}{N} \sum_n \mathbb{E}_{z \sim q(z|x_n)}[\log \pi(x_n; g(z))] - \text{KL}(q(z \mid x_n) \,\|\, p(z)) \leq \frac{1}{N} \sum_n \log p(x_n), \quad (1)$$

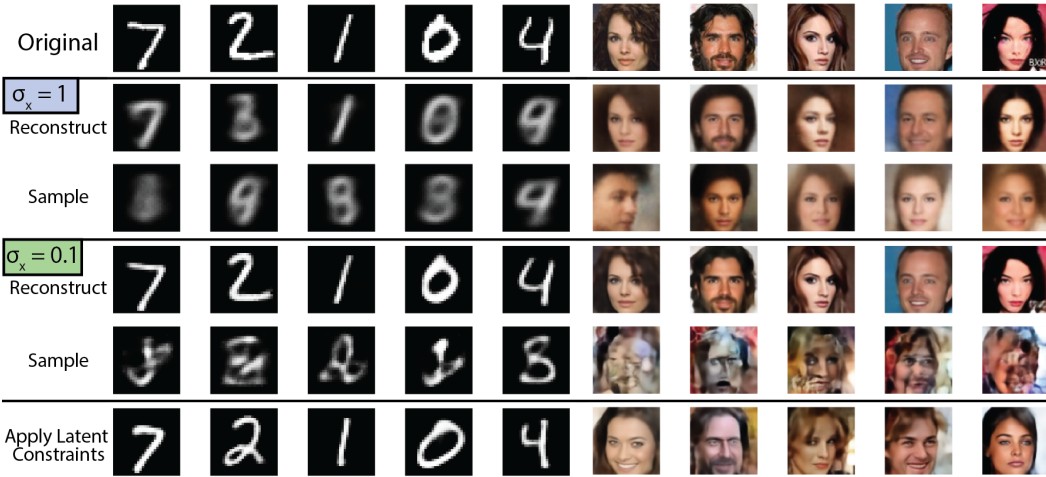

Figure 2: Typical VAEs use a pixel-wise data likelihood, $\mathcal{N}(\mu_x(z), \sigma_x I)$, with $\sigma_x = 1$ to produce coherent samples at the expense of visual and conceptual blurriness (Row 3). Some reconstructions (Row 2) actually change attributes of the original data. Decreasing $\sigma_x$ to 0.1 maximizes the ELBO (supplemental Table 4) and increases the fidelity of reconstructions (Row 4) at the cost of sample realism (Row 5). Using an actor to shift prior samples to satisfy the realism constraint, we achieve more realistic samples without sacrificing sharpness (Row 6). The samples are mapped to the closest point in latent space that both satisfies the realism constraint and has the same attributes as the original data.

where the "encoder" distribution $q(z \mid x)$ is an approximation to the posterior $p(z \mid x)$, $\pi(x; g(z)) \triangleq p(x \mid z)$ is a tractable likelihood function that depends on some parameters output by a "decoder" function $g(z)$, and $q$ and $g$ are fit to maximize the evidence lower bound (ELBO) $\mathcal{L}^{\text{ELBO}}$. The likelihood $\pi(x; g)$ is often chosen to be a product of simple distributions such as $\pi(x; g) = \mathcal{N}(x; g, \sigma_x^2 I)$ for continuous data or $\pi(x; g) = \prod_d \text{Bernoulli}(x_d; g_d)$ for binary data.

GANs and VAEs have complementary strengths and weaknesses. GANs suffer from the "mode-collapse" problem, where the generator assigns mass to a small subset of the support of the population distribution—that is, it may generate realistic samples, but there are many more realistic samples that it cannot generate. This is particularly problematic if we want to use GANs to *manipulate* data rather than generate new data; even GAN variants that include some kind of inference machinery (e.g., Donahue et al., 2016; Dumoulin et al., 2016; Perarnau et al., 2016) to determine what $z$ best matches some $x$ tend to produce reconstructions that are reminiscent of the input but do not preserve its identity.

On the other hand, VAEs (especially those with simple likelihoods $\pi$) often exhibit a tradeoff between sharp reconstructions and sensible-looking samples (see Figure 2). That is, depending on what hyperparameters they are trained with (e.g., latent dimensionality and the scale of the likelihood term), VAEs tend to either produce blurry reconstructions and plausible (but blurry) novel samples, or bizarre samples but sharp reconstructions. It has been argued (Makhzani et al., 2016) that this is due to the "holes" problem; the decoder is trained on samples from the marginal posterior $q(z) \triangleq \frac{1}{N} \sum_n q(z \mid x_n)$, which may have very high KL divergence to the presupposed marginal $p(z)$ (Hoffman & Johnson, 2016). In particular, if the decoder, $g(z)$, can reconstruct arbitrary values of $x$ with high accuracy (as in the case of small $\sigma_x$) then the typical posterior $p(z \mid x)$ will be highly concentrated. We show this experimentally in supplemental Figure 16. If $q(z \mid x)$ underestimates the posterior variance (as it usually does), then the marginal posterior $q(z)$ will also be highly concentrated, and samples from $p(x) = \int_z p(z)p(x \mid z)dz$ may produce results that are far from typical reconstructions $\mathbb{E}_p[x \mid z \sim q(z \mid x)]$. If we tune $\sigma_x$ to maximize the ELBO (Bishop, 2006), we find the optimal $\sigma_x \approx 0.1$ (supplemental Table 4). Figure 2 shows that this choice does indeed lead to good reconstructions but strange-looking samples.

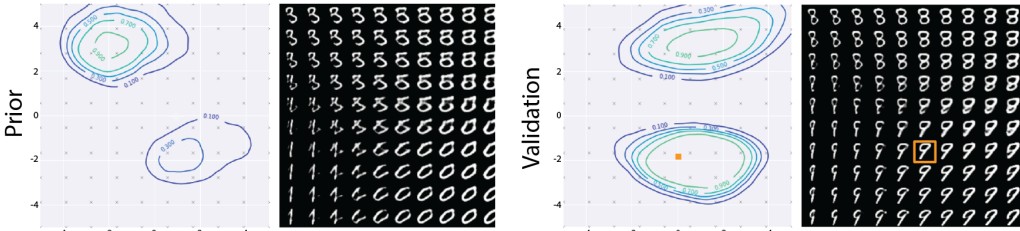

Figure 3: Contour maps of the critic value functions for the marginal posterior ("realism") constraint. We look at the two latent dimensions that have the lowest average posterior standard deviation on the training set, taking low variance in $z$ space as a proxy for influence over the generated images. All other latent dimensions are held fixed at their original values (from a sample from $p(z)$ on the left, and from a sample from $q(z \mid x)$ for a held-out $x$ on the right). Gray x marks correspond to the points in latent space of the generated images to the right. The cross-section on the left, taken from a prior sample, shows contours that point towards more realistic looking digits. In the cross-section on the right, a sample from the validation set (indicated by orange squares) resides within a local maximum of the critic, as one would hope.

Conditional GANs (CGAN; Mirza & Osindero, 2014) and conditional VAEs (CVAE; Sohn et al., 2015) can generate samples conditioned on attribute information when available, but they must be trained with knowledge of the attribute labels for the whole training set, and it is not clear how to adapt them to new attributes without retraining from scratch. Furthermore, CGANs and CVAEs suffer from the same problems of mode-collapse and blurriness as their unconditional cousins.

We take a different approach to conditional generation and identity-preserving transformation. We begin by training an unconditional VAE with hyperparameters chosen to ensure good reconstruction (at the expense of sample quality). We then train a "realism" critic to predict whether a given $z$ maps to a high-quality sample. We also train critics to predict whether a given $z$ maps to a sample that manifests various attributes of interest. To generate samples that are both realistic and exhibit desired attributes, one option is to optimize random $z$ vectors until they satisfy both the realism and attribute critics. Alternately, we can amortize this cost by training an "actor" network to map a random set of $z$ vectors to a subregion of latent space that satisfies the constraints encoded by the critics. By encouraging these transformed $z$ vectors to remain as close as possible to where they started, we alleviate the mode-collapse problem common to GANs.

Our approach is summarized visually in Figure 1. The details follow in sections 3, 4, 5, and 6.

## 3 THE "REALISM" CONSTRAINT: SHARPENING VAE SAMPLES

We define the realism constraint implicitly as being satisfied by samples from the marginal posterior $q(z) \triangleq \frac{1}{N} \sum_n q(z \mid x_n)$ and not those from $p(z)$. By enforcing this constraint, we can close the gap between reconstruction quality and sample quality (without sacrificing sample diversity).

As shown in Figure 1, we can train a critic $D$ to differentiate between samples from $p(z)$ and $q(z)$. The critic loss, $\mathcal{L}_D(z)$, is simply the cross-entropy, with labels $c = 1$ for $z \sim q(z \mid x)$ and $c = 0$ for $z \sim p(z)$. We found that the realism critic had little trouble generalizing to unseen data; that is, it was able to recognize samples from $q(z \mid x^{\text{held-out}})$ as being "realistic" (Figure 3).

Sampling from the prior is sufficient to train $D$ for models with lower KL Divergence, but if the KL Divergence between $q$ and $p$ is large, the chances of sampling a point $p(z)$ that has high probability under $q(z)$ becomes vanishingly small. This leads to poor sample quality and makes it difficult for $D$ to learn a tight approximation of $q(z)$ solely by sampling from $p(z)$. Instead, we use an inner-loop of gradient-based optimization, $G_{\text{opt}}(z) = \text{GradientDescent}(z; \mathcal{L}_D(z))$, to move prior samples to points deemed more like $q(z)$ by $D$. For clarity, we introduce the shorthand $\mathcal{L}_{c=1}(z) \triangleq -\log(D(z))$ and $\mathcal{L}_{c=0}(z) \triangleq -(1 - \log(D(z)))$. This gives us our critic loss for the realism constraint:

$$\mathcal{L}_D(z) = \mathbb{E}_{z \sim q(z|x)}[\mathcal{L}_{c=1}(z)] + \mathbb{E}_{z \sim p(z)}[\mathcal{L}_{c=0}(z)] + \mathbb{E}_{z \sim G(p(z))}[\mathcal{L}_{c=0}(z)] \tag{2}$$

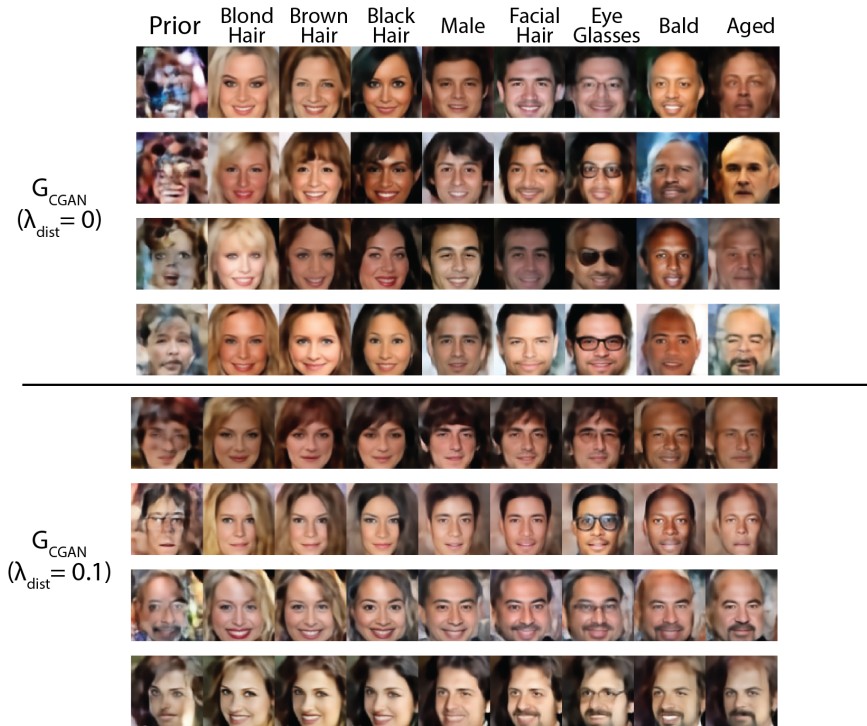

Figure 4: Conditional generation with a CGAN actor-critic pair acting in the latent space of a VAE with $\sigma_x = 0.1$. Each row starts from a different prior sample and maps it to a new point in latent space that satisfies both the attribute constraints and the realism constraint. The attribute constraints are changed one at a time to produce as smooth a transition as possible from left to right. The bottom CGAN is regularized during training to prefer small shifts in latent space ($\lambda_{\text{dist}} = 0.1$), while the top is not ($\lambda_{\text{dist}} = 0.0$). Compared to the images generated by the unregularized model, the images generated by the regularized model are much less diverse across columns, suggesting that the regularization does indeed enforce some degree of identity preservation. The regularized model produces images that are somewhat *more* diverse across rows, suggesting that the regularization fights mode collapse (arguably at the expense of image quality). For each column, the complete list of attributes is given in supplemental Table 3.

Since this inner-loop of optimization can slow down training, we amortize the generation by using a neural network as a function approximator. There are many examples of such amortization tricks, including the encoder of a VAE, generator of a GAN, and fast neural style transfer (Ulyanov et al., 2016; Li & Wand, 2016; Johnson et al., 2016). As with a traditional GAN, the parameters of the function $G$ are updated to maximize the value $D$ ascribes to the shifted latent points. One of the challenges using a GAN in this situation is that it is prone to mode-collapse. However, an advantage of applying the GAN in latent space is that we can regularize $G$ to try and find the closest point in latent space that satisfies $D$, thus encouraging diverse solutions. We introduce a regularization term, $\mathcal{L}_{\text{dist}}(z', z) = 1/\bar{\sigma}_z{}^2 \log(1 + (z' - z)^2)$ to encourage nearby solutions, while allowing more exploration than a mean square error term. As a VAE utilizes only a fraction of its latent dimensions, we scale the distance penalty of each dimension by its utilization, as indicated by the squared reciprocal of the scale $\sigma_z(x)$ of the encoder distribution $q(z \mid x)$, averaged over the training dataset, $\bar{\sigma}_z \triangleq \frac{1}{N} \sum_n \sigma_z(x_n)$. The regularized loss is

$$\mathcal{L}_G(z) = \mathbb{E}_{z \sim p(z)}[\mathcal{L}_{c=1}(G(z)) + \lambda_{\text{dist}} \mathcal{L}_{\text{dist}}(G(z), z)]. \tag{3}$$

## 4 ATTRIBUTE CONSTRAINTS: CONDITIONAL GENERATION

We want to generate samples that are realistic, but we also want to control what attributes they exhibit. Given binary attribute labels $y$ for a dataset, we can accomplish this by using a CGAN in

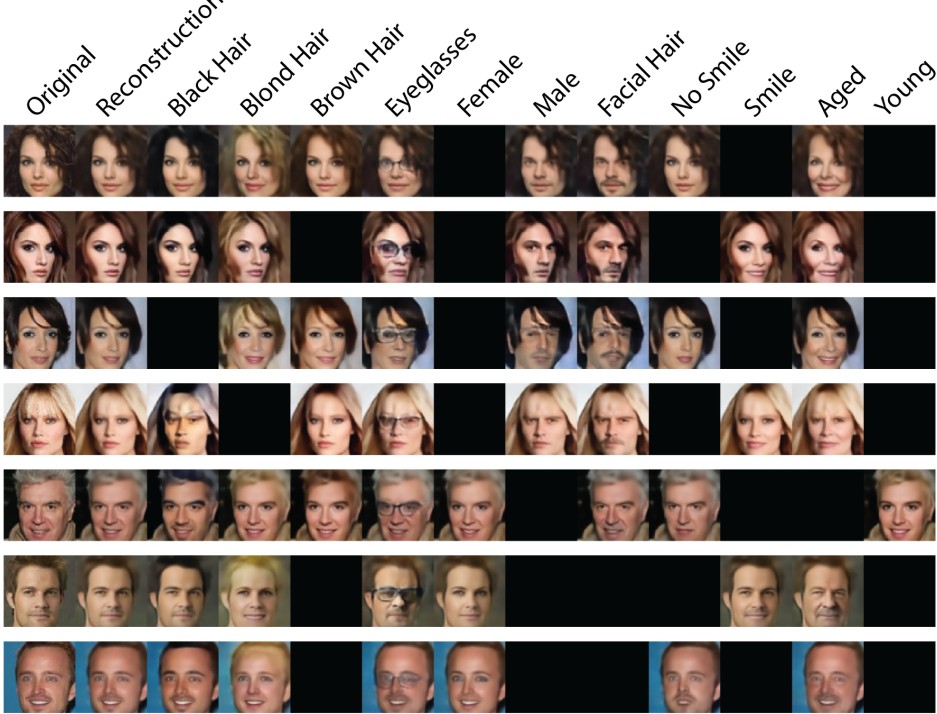

Figure 5: Identity-preserving transformations with optimization. Two separate critics are trained, one for attributes and one for the realism constraint. Starting at the latent points corresponding to the data reconstructions, we then perform gradient ascent in latent space on a weighted combination of critic values (1.0 attribute, 0.1 marginal posterior), stopping when a threshold value is passed for both critics. Images remain semantically close to the original because the pixel-wise likelihood of VAE training encourages identity-preserving reconstructions, and the dynamics of gradient ascent are naturally limited to finding solutions close in latent space. Panels are black for attributes of the original image, as the procedure just returns the original point in latent space.

the latent space, which amounts to replacing $D(z)$ and $G(z)$ with conditional versions $D(z, y)$ and $G(z, y)$ and concatenating $y$ to $z$ as input. If both the actor and critic see attribute information, $G$ must find points in latent space that could be samples from $q(z)$ with attributes $y$.

This procedure is computationally inexpensive relative to training a generative model from scratch. In most of our experiments, we use a relatively large CGAN actor-critic pair (4 fully connected ReLU layers of 2048 units each), which during training uses about $96\times$ fewer FLOPs/iteration than the unconditional VAE. We also trained a much smaller CGAN actor-critic pair (3 fully connected ReLU layers of 256 units), which uses about $2884\times$ fewer FLOPs/iteration than the VAE, and achieves only slightly worse results than the larger CGAN (supplemental Figure 14 and Table 1).

Figure 4 demonstrates the quality of conditional samples from a CGAN actor-critic pair and the effect of the distance penalty, which constrains generation to be closer to the prior sample, maintaining similarity between samples with different attributes. The regularized CGAN actor has less freedom to ignore modes by pushing many random $z$ vectors to the same area of the latent space, since it is penalized for moving samples from $p(z)$ too far. The increased diversity across rows of the regularized CGAN is evidence that this regularization does fight mode-collapse (additional qualitative evidence is in supplemental Figures 7 and 8). However, without a distance penalty, samples appear more a bit realistic with more prominent attributes. This is supported by Table 1, where we use a separately trained attribute classification model to quantitatively evaluate samples. The actor with no penalty generates samples that are more accurately classified than the actor with a penalty but also shifts the samples much farther in latent space.

| CelebA | Accuracy | Precision | Recall | F1 Score | $z_{\mathrm{MSE}}$ |
|---|---|---|---|---|---|
| (This Work) 10 Attributes | | | | | |
| Test Data | 0.936 | 0.901 | 0.893 | 0.895 | |
| $G_{CGAN}(\lambda_{\mathrm{dist}} = 0)$ | 0.942 | 0.914 | 0.904 | 0.906 | 80.7 |
| $G_{CGAN}(\lambda_{\mathrm{dist}} = 0)$ (Small Model) | 0.926 | 0.898 | 0.860 | 0.870 | 58.9 |
| $G_{CGAN}(\lambda_{\mathrm{dist}} = 0.1)$ | 0.928 | 0.903 | 0.863 | 0.874 | 17.0 |
| (Perarnau et al., 2016) 18 Attributes | | | | | |
| Test Data | 0.928 | | | 0.715 | |
| IcGAN | 0.860 | | | 0.524 | |

Table 1: Accuracy of a separate model trained to classify attributes from images, evaluated on test data and generated images. We condition and evaluate the generated images on the same labels as the test data. For comparison, the results of a similar task using invertible CGANs for generation (Perarnau et al., 2016) are provided. However, since the full list of salient attributes was not given in the paper, we emphasize that they are not directly comparable as the two experiments use a slightly different set of attribute labels. We also measure the distance in latent space that prior samples are shifted, weighted by $1/\bar{\sigma}_z{}^2$. Actors trained with a latent distance penalty $\lambda_{\mathrm{dist}}$ have slightly worse accuracy, but find latent points much closer to the prior samples and produce a greater diversity of images (see supplemental Figures 7 and 8). Interestingly, an actor trained without a distance penalty achieves higher classification accuracy than the test set itself, possibly by generating images with more exaggerated and distinctive features than real data. A "small model" CGAN with 85x fewer parameters (3 fully connected layers of 256 units) generates images (supplemental Figure 14) of comperable quality. Due to the smaller capacity, the model finds more local solutions (smaller $z_{MSE}$) that have slightly less attribute accuracy, but are more visually similar to the prior sample without an explicit regularization term.

Although we used a VAE as the base generative model, our approach could also be used to generate high-quality conditional samples from pretrained classical autoencoders. We show in supplemental Figure 15 that we obtain reasonably good conditional samples (albeit with high-frequency spatial artifacts) as $\sigma_x \to 0$ (equivalent to a classical autoencoder). Learning the decoder using VAE training encourages q(z) to fill up as much of the latent space as possible (without sacrificing reconstruction quality), which in turn encourages the decoder to map more of the latent space to reasonable-looking images. The prior $p(z) = \mathcal{N}(0, I)$ also imposes a natural scale on the latent variables.

## 5 IDENTITY-PRESERVING TRANSFORMATIONS

If we have a VAE that can produce good reconstructions of held-out data, we can transform the attributes of the output by gradient-based optimization. We simply need to train a critic, $D_{attr}(z)$, to predict the attribute labels $p(y \mid z)$ of the data embeddings $z \sim q(z \mid x)$, and use a cross-entropy loss to train. Then, starting from a data point, $z \sim q(z \mid x)$, we can perform gradient descent on the the realism constraint and attribute constraint jointly, $\mathcal{L}_{D_{\mathrm{real}}}(z) + \lambda_{\mathrm{attr}}\mathcal{L}_{D_{\mathrm{attr}}}(z)$. Note that it is helpful to maintain the realism constraint to keep the image from distorting unrealistically. Using the same procedure, we can also conditionally generate new samples (supplemental Figure 9) by starting from $z \sim p(z)$.

Figure 5 demonstrates transformations applied to samples from the held-out evaluation dataset. Note that since the reconstructions are close to the original images, the transformed images also maintain much of their structure. This contrasts with supplemental Figure 10, where a distance-penalty-free CGAN actor produces transformations that share attributes with the original but shift identity. We could preserve identity by introducing a distance penalty, but find that it is much easier to find the correct weighting of realism cost, attribute cost, and distance penalty through optimization, as each combination does not require retraining the network.

## 6 RULE-BASED CONSTRAINTS: ZERO-SHOT CONDITIONAL GENERATION

So far, we have assumed access to labeled data to train attribute classifiers. We can remove the need to provide labeled examples by leveraging the structure learned by our pre-trained model, using it to generate exemplars that are scored by a user-supplied reward function. If we constrain the reward

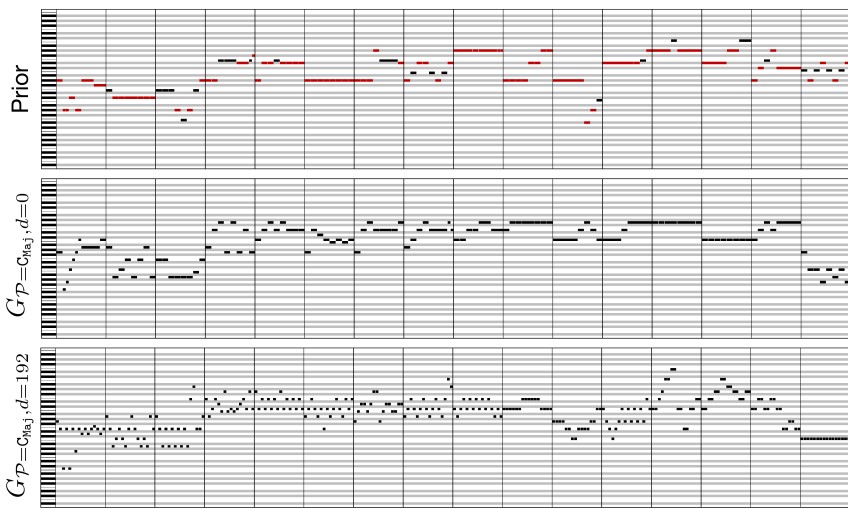

Figure 6: Transformations from a prior sample for the Melody VAE model. In each 16-bar pianoroll, time is in the horizontal direction and pitch in the vertical direction. In the prior sample, notes falling outside of the C Major scale are shown in red. After transformation by $G_{\mathcal{P}=\mathrm{C_{Maj}},d=0}$, all sampled notes fall within the scale, without a significant change to note density. After transformation of the original $z$ by $G_{\mathcal{P}=\mathrm{C_{Maj}},d=192}$, all sampled notes lay within the scale and the density increases beyond 192. Synthesized audio of these samples can be heard at https://goo.gl/ouULt9.

function to be bounded, $c(x) : \mathbb{R}^N \to [0,1]$, the problem becomes very similar to previous GAN settings, but now the actor, $G$, and critic, $D$, are working together. $D$ aims to best approximate the true value of each latent state, $\mathbb{E}_{x \sim p(x|z)}\, c(x)$, and $G$ aims to shift samples from the prior to high-value states. The critic loss is the cross-entropy from $c(x)$, and the actor loss is the same as $\mathcal{L}_G$ in equation 3, where we again have a distance penalty to promote diversity of outputs.

Note that the reward function and VAE decoder need not necessarily be differentiable, as the critic learns a value function to approximate the reward, which the actor uses for training. To highlight this, we demonstrate that the output of a recurrent VAE model can be constrained to satisfy hard-coded rule-based constraints.

We first train an LSTM VAE (details in the Appendix) on melodic fragments. Each melody, $m$, is represented as a sequence of categorical variables. In order to examine our ability to constrain the pitch classes and note density of the outputs, we define two reward functions, one that encourages notes from a set of pitches $\mathcal{P}$, and another for that encourages melodies to have at least $d$ notes:

$$c_{\mathrm{pitch}}(m, \mathcal{P}) = \sum_{p \in m} \mathbb{1}(p \in \mathcal{P})/|m| \qquad c_{\mathrm{density}}(m, d) = \min(1, |m|/d) \qquad (4)$$

Figure 6 gives an example of controlling the pitch class and note density of generated outputs, which is quantitatively supported by the results in Table 2. During training, the actor goes through several phases of exploration and exploitation, oscillating between expanding to find new modes with high reward and then contracting to find the nearest locations of those modes, eventually settling into high value states that require only small movements in the latent space (supplemental Figure 11).

## 7 RELATED WORK

Conditional GANs (Mirza & Osindero, 2014) and VAEs (Sohn et al., 2015) introduce conditioning variables at training time. Sohn et al. (2015) allow these variables to affect the distribution in latent $z$ space, but still require that $p(z \mid y)$ be a tractable distribution. Perarnau et al. (2016) use CGANs to adjust images, but because CGANs cannot usually reconstruct arbitrary inputs accurately, they must resort to image-space processing techniques to transfer effects to the original input. White (2016) propose adding "attribute vectors" to samples from $p(z)$ as a simple and effective heuristic to perform transformations, which relies heavily on the linearity of the latent space.

| Actor | $c_{\text{pitch}}(m, \mathcal{P} = \text{C}_{\text{Maj}})$ | $c_{\text{density}}(m, d = 192)$ | $z_{\text{MSE}}$ |
|---|---|---|---|
| Prior | 0.579 (0.43%) | 0.417 (0.04%) | - |
| $G_{\mathcal{P}=\text{C}_{\text{Maj}}, d=0}$ | 0.991 (70.8%) | 0.459 (0.01%) | 0.015 |
| $G_{\mathcal{P}=\text{C}_{\text{Maj}}, d=192}$ | 0.982 (62.4%) | 0.985 (84.9%) | 0.039 |

Table 2: Average rewards and constraint satisfaction rates (in parentheses) for unconditional (Prior) and conditional generation. Samples from the prior receive low rewards, on average, and near zero satisfaction rates from both the pitch class (C Major) and note density ($\geq$ 192 notes) constraints. After applying an actor optimized only for the C Major scale ($G_{\mathcal{P}=\text{C}_{\text{Maj}}, d=0}$), the pitch class constraint is fully satisfied 70.8% of the time with only a minor effect on density. The average value close to 1 also indicates that when the constraint is not satisfied, it is typically off by only a few notes. Applying an actor function optimized for the C Major scale and high density ($G_{\mathcal{P}=\text{C}_{\text{Maj}}, d=192}$) causes both constraints to be satisfied at high rates, with a slightly larger shift in latent space.

Some recent work has focused on applying more expressive prior constraints to VAEs (Rezende et al., 2014; Sønderby et al., 2016; Chen et al., 2017; Tomczak & Welling, 2017). The prior that maximizes the ELBO is $p^\star(z) = q(z)$ (Hoffman & Johnson, 2016); one can interpret our realism constraint as trying to find an implicit distribution that is indistinguishable from $q(z)$. Like the adversarial autoencoder of Makhzani et al. (2016), our realism constraint relies on a discriminative model, but instead of trying to force $q(z)$ to equal some simple $p(z)$, we only weakly constrain $q(z)$ and then use a classifier to "clean up" our results.

Like this work, the recently proposed adversarially regularized autoencoder (Junbo et al., 2017) uses adversarial training to generate latent codes in a latent space discovered by an autoencoder; that work focuses on unconditional generation. Gómez-Bombarelli et al. (2016) train classifiers in the latent space of a VAE to predict what latent variables map to molecules with various properties, and then use iterative gradient-based optimization in the latent space to find molecules that have a desired set of properties. On molecule data, their procedure generates invalid molecules rarely enough that they can simply reject these samples, which are detected using off-the-shelf software. By contrast, the probability of generating realistic images under our pretrained VAE is astronomically small, and no simple criterion for detecting valid images exists.

Jaques et al. (2017) also use a classifier to constrain generation; they use a Deep Q-network as an auxiliary loss for training an LSTM. Closest to Section 6, Nguyen et al. (2016a;b) generate very high quality conditional images by optimizing a sample from the latent space of a generative network to create an image that maximizes the class activations of a pretrained ImageNet classifier. Our work differs in that we learn an amortized generator/discriminator directly in the latent space and we achieve diversity through regularizing by the natural scale of the latent space rather than through a modified Langevin sampling algorithm.

## 8 DISCUSSION AND FUTURE WORK

We have demonstrated a new approach to conditional generation by constraining the latent space of an unconditional generative model. This approach could be extended in a number of ways.

One possibility would be to plug in different architectures, including powerful autoregressive decoders or adversarial decoder costs, as we make no assumptions specific to independent likelihoods. While we have considered constraints based on implicit density estimation, we could also estimate the constrained distribution directly with an explicit autoregressive model or another variational autoencoder. The efficacy of autoregressive priors in VAEs is promising for this approach (Kingma et al., 2016). Conditional samples could then be obtained by ancestral sampling, and transformations by using gradient ascent to increase the likelihood under the model. Active or semisupervised learning approaches could reduce the sample complexity of learning constraints. Real-time constraint learning would also enable new applications; it might be fruitful to extend the reward approximation of Section 6 to incorporate user preferences as in (Christiano et al., 2017).

ACKNOWLEDGMENTS

Many thanks to Jascha Sohl-Dickstein, Colin Raffel, and Doug Eck for their helpful brainstorming and encouragement.

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

# 9 APPENDIX

## 9.1 EXPERIMENTAL DETAILS

For images, we use the MNIST digits dataset (LeCun & Cortes, 2010) and the Large-scale Celeb-Faces Attributes (CelebA) dataset (Liu et al., 2015). MNIST images are $28 \times 28$ pixels and greyscale scaled to [0, 1]. For attributes, we use the number class label of each digit. CelebA images are center-cropped to $128 \times 128$ pixels and then downsampled to $64 \times 64$ RGB pixels and scaled to [0, 1]. We find that many of the attribute labels are not strongly correlated with changes in the images, so we narrow the original 40 attributes to the 10 most visually salient: blond hair, black hair, brown hair, bald, eyeglasses, facial hair, hat, smiling, gender, and age.

For melodies, we scraped the web to collect over 1.5 million publicly available MIDI files. We then extracted 16-bar melodies by sliding a window with a single bar stride over each non-percussion instrument with a $\frac{4}{4}$ time signature, keeping only the note with the highest pitch when multiple overlap. This produced over 3 million unique melodies. We represent each melody as a sequence of 256 (16 per bar) categorical variables taking one of 130 discrete states at each sixteenth note: 128 note-on pitches, a hold state, and a rest state.

## 9.2 MODEL ARCHITECTURES

All encoders, decoders, and classifiers are trained with the Adam optimizer (Kingma & Ba, 2015), with learning rate = 3e-4, $\beta_1 = 0.9$, and $\beta_2 = 0.999$.

To train $D_{real}(z)$, $D_{attr}(z)$ and $G(z)$ we follow the training procedure of Gulrajani et al. (2017), applying a gradient penalty of 10, training $D$ and $G$ in a 10:1 step ratio, and use the Adam optimizer with learning rate = 3e-4, $\beta_1 = 0.0$, and $\beta_2 = 0.9$. While not necessary to converge, we find it improves the stability of optimization. We do not apply any of the other tricks of GAN training such as batch normalization, minibatch discrimination, or one-sided label smoothing (Radford et al., 2015; Salimans et al., 2016). As samples from $p(z)$ are easier to discriminate than samples from $G(p(z))$, we train $D$ by sampling from $p(z)$ at a rate 10 times less than $G(p(z))$. For actors with inner-loop optimization, $G_{\text{opt}}$, 100 iterations of Adam are used with with learning rate = 1e-1, $\beta_1 = 0.9$, and $\beta_2 = 0.999$.

### 9.2.1 MNIST FEED-FORWARD VAE

To model the MNIST data, we use a deep feed-forward neural network (Figure 13a).

The encoder is a series of 3 linear layers with 1024 outputs, each followed by a ReLU, after which an additional linear layer is used to produce 2048 outputs. Half of the outputs are used as the $\mu$ and the softplus of the other half are used as the $\sigma$ to parameterize a 1024-dimension multivariate Gaussian distribution with a diagonal covariance matrix for $z$.

The decoder is a series of 3 linear layers with 1024 outputs, each followed by a ReLU, after which an additional linear layer is used to produce 28x28 outputs. These outputs are then passed through a sigmoid to generate the output image.

### 9.2.2 CELEBA CONVOLUTIONAL VAE

To model the CelebA data, we use a deep convolutional neural network (Figure 13b).

The encoder is a series of 4 2D convolutional layers, each followed by a ReLU. The convolution kernels are of size $3 \times 3$, $3 \times 3$, $5 \times 5$, and $5 \times 5$, with 2048, 1024, 512, and 256 output channels, respectively. All convolutional layers have a stride of 2. After the final ReLU, a linear layer is used to produce 2048 outputs. Half of the outputs are used as the $\mu$ and the softplus of the other half are used as the $\sigma$ to parameterize a 1024-dimension multivariate Gaussian distribution with a diagonal covariance matrix for $z$.

The decoder passes the $z$ through a 4x4x2048 linear layer, and then a series of 4 2D transposed convolutional layers, all but the last of which are followed by a ReLU. The deconvolution kernels are of size $5 \times 5$, $5 \times 5$, $3 \times 3$, and $3 \times 3$, with 1024, 512, 256, and 3 output channels, respectively. All

deconvolution layers have a stride of 2. The output from the final deconvolution is passed through a sigmoid to generate the output image.

The classifier that is trained to predict labels from images are identical to the VAE encoders except that they end with a sigmoid cross-entropy loss.

### 9.2.3 MELODY SEQUENCE VAE

Music is fundamentally sequential, so we use an LSTM-based sequence VAE for modelling mono-phonic melodies (Figure 13c).

The encoder is made up of a single-layer bidirectional LSTM, with 2048 units per cell. The final output in each direction is concatenated and passed through a linear layer to produce 1024 outputs. Half of the outputs are used as the $\mu$ and the softplus of the other half are used as a $\sigma$ to parameterize a 512-dimension multivariate Gaussian distribution with a diagonal covariance matrix for $z$.

Since musical sequences often have structure at the bar level, we use a hierarchical decoder to model long melodies. First, the $z$ goes through a linear layer to initialize the state of a 2-layer LSTM with 1024 units per layer, which outputs 16 embeddings of size 512 each, one per bar. Each of these embeddings are passed through a linear layer to produce 16 initial states for another 2-layer LSTM with 1024 units per layer. This bar-level LSTM autoregressively produces individual sixteenth note events, passing its output through a linear layer and softmax to create a distribution over the 130 classes. This categorical distribution is used to compute a cross-entropy loss during training or samples at inference time. In addition to generating the initial state at the start of each bar, the embedding for the current bar is concatenated with the previous output as the input at each time step.

### 9.2.4 ACTOR FEED-FORWARD NETWORK

For $G(z)$, we use a deep feed-forward neural network (Figure 12a) in all of our experiments.

The network is a series of 4 linear layers with 2048 outputs, each followed by a ReLU, after which an additional linear layer is used to produce $2 * dim(z)$ outputs. Half of the outputs are used as the $\delta z$ and the sigmoid of the other half are used as $gates$. The transformed $z'$ is the computed as $(1 - gates) * z + gates * \delta z$. This aids in training as the network only has to then predict shifts in $z$.

When conditioning on attribute labels, $y$, to compute $G(z, y)$, the labels are passed through a linear layer producing 2048 outputs which are concatenated with $z$ as the model input.

### 9.2.5 CRITIC FEED-FORWARD NETWORK

For $D(z)$, we use a deep feed-forward neural network (Figure 12b) in all of our experiments.

The network is a series of 4 linear layers with 2048 outputs, each followed by a ReLU, after which an additional linear layer is used to produce a single output. This output is passed through a sigmoid to compute $D(z)$.

When conditioning on attribute labels, $y$, to compute $D(z, y)$, the labels are passed through a linear layer producing 2048 outputs which are concatenated with $z$ as the model input.

## 9.3 SUPPLEMENTAL FIGURES

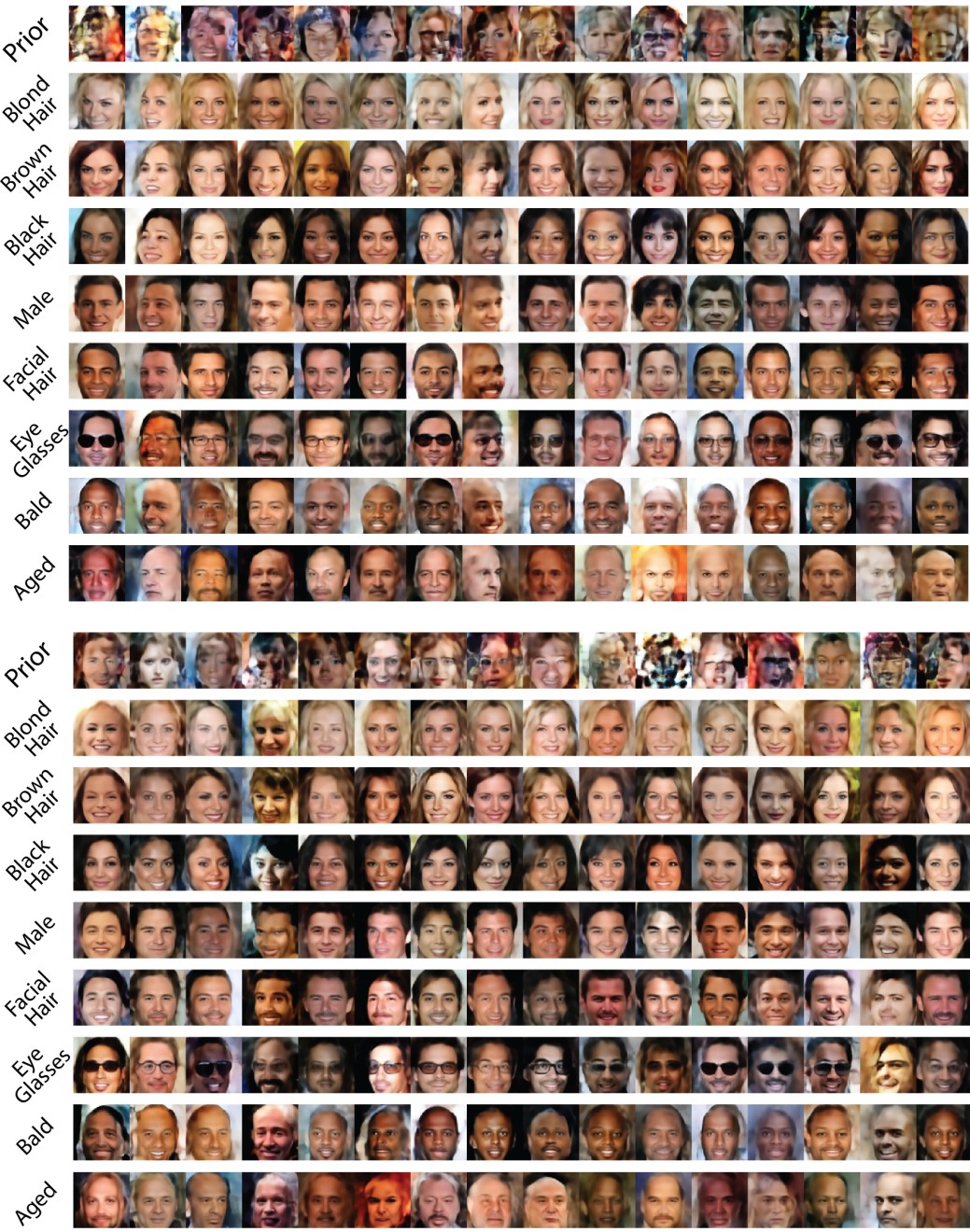

Figure 7: Additional generated CelebA faces by $G_{\text{CGAN}}$ with $\lambda_{\text{dist}} = 0$. Full attribute labels are given in supplementary Table 3

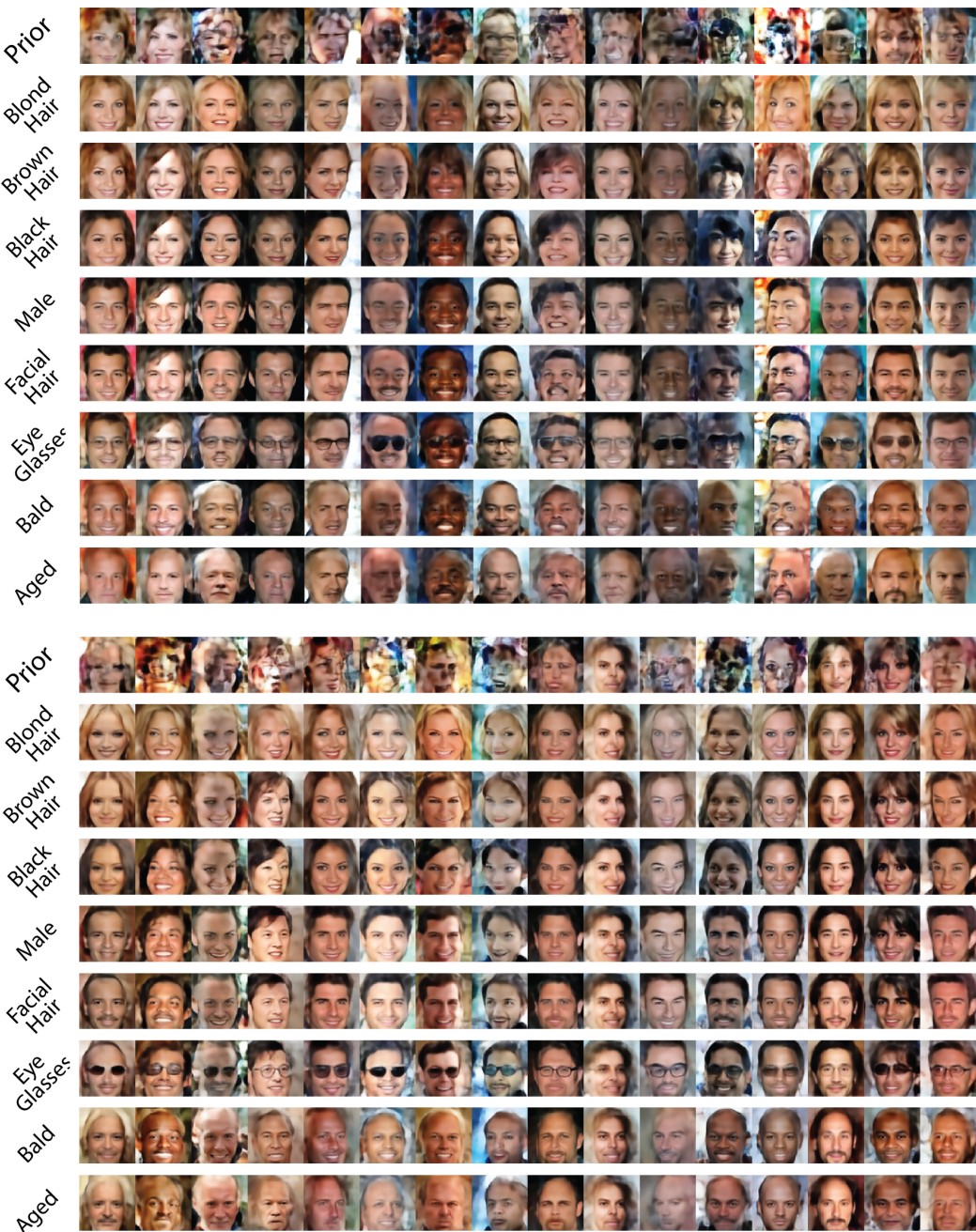

Figure 8: Additional generated CelebA faces by $G_{\mathrm{CGAN}}$ with $\lambda_{\mathrm{dist}} = 0.1$. Full attribute labels are given in supplementary Table 3

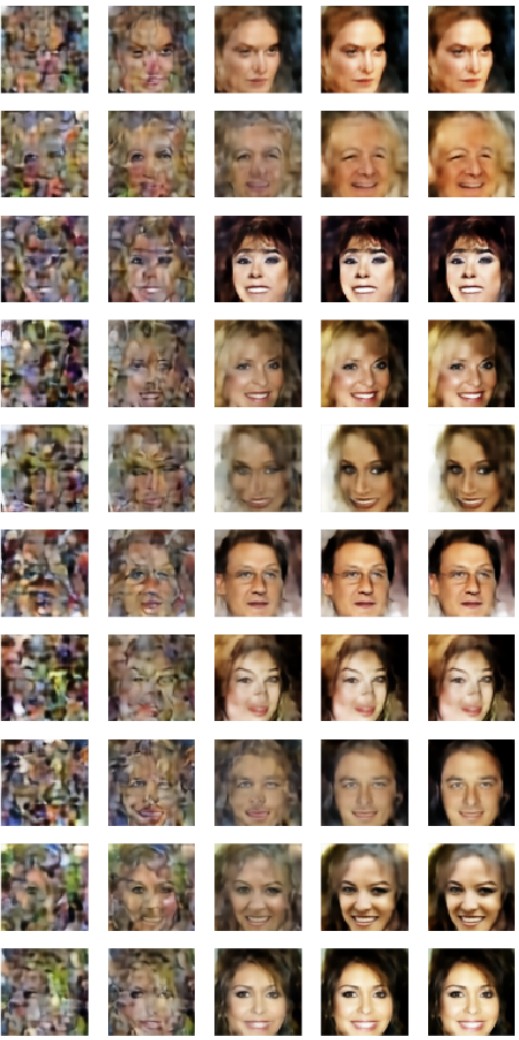

Figure 9: Optimization of samples drawn from the prior to satisfy both the realism constraint and attribute constraints (drawn from the test set). The optimization takes 100 steps, and images are shown at 0, 10, 30, 50 and 100 steps. $D$ is trained with inner-loop optimization, $G_{opt}$, as described in Section 9.2

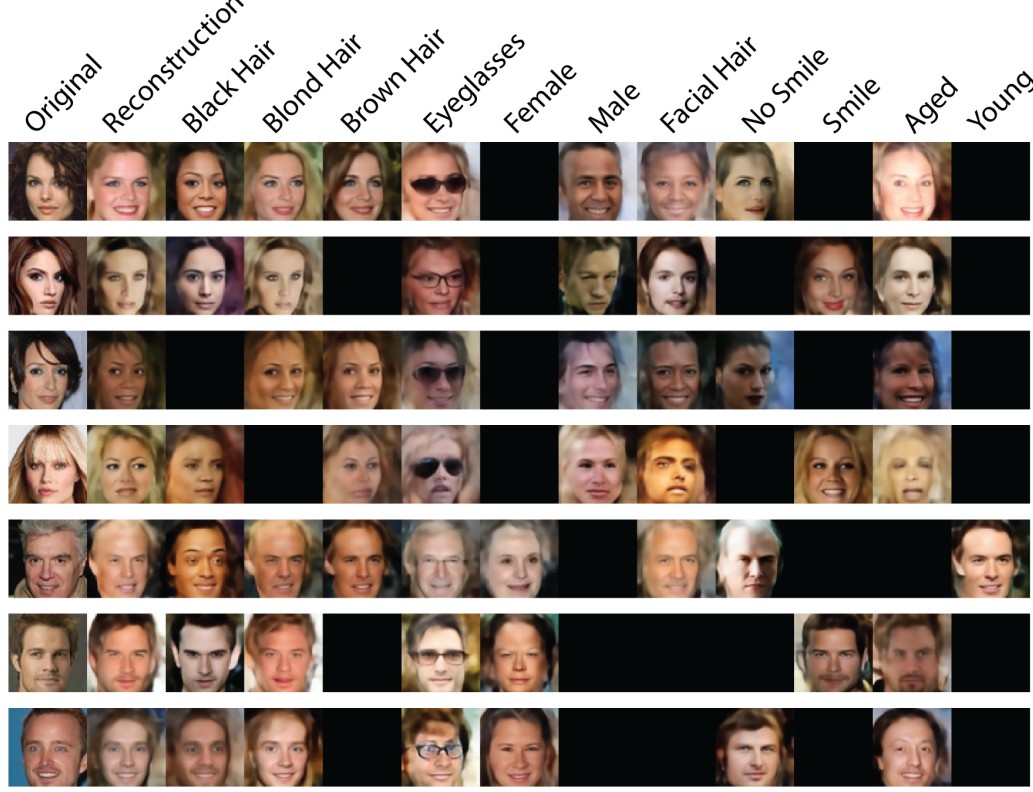

Figure 10: Identity-distorting transformations with CGAN actor-critic. Without a penalty to encourage small moves in latent space, the actor maps the latent vectors of the original data points to generated images that have the correct attributes, but a different identity. Panels are black for attributes of the original image, as the procedure just returns the same image as the reconstruction.

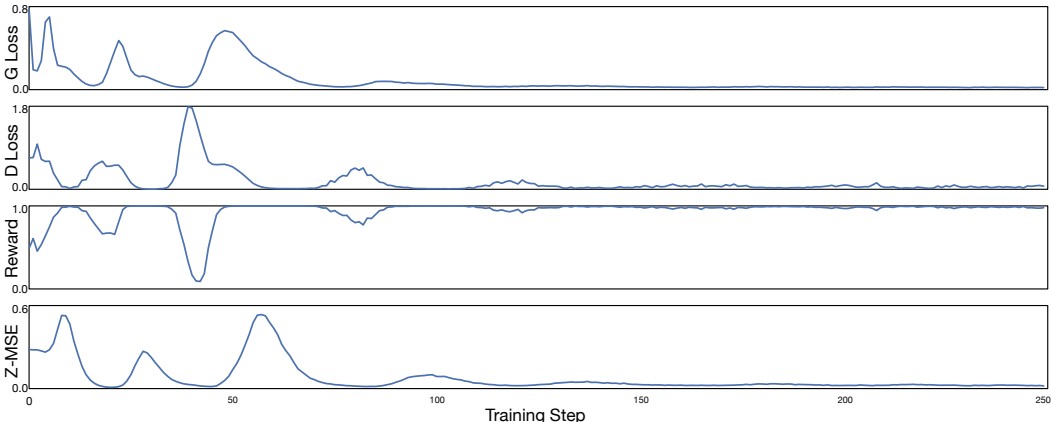

Figure 11: Training curves for melody actor ($G$) and critic ($D$) pair for pitch class constraint $c_{\text{pitch}}(m, \mathcal{P} = \text{C}_{\text{Maj}})$.

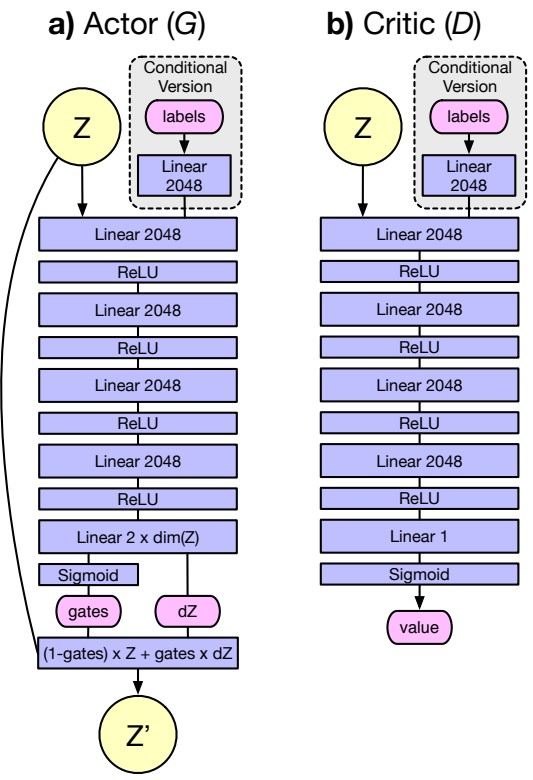

Figure 12: Architecture for the (a) actors and (b) critics used in all experiments.

| Figure Label | Bald | Black Hair | Blond Hair | Brown Hair | Eye-glasses | Male | Beard | Smiling | Hat | Young |
|---|---|---|---|---|---|---|---|---|---|---|
| Blond Hair | 0 | 0 | 1 | 0 | 0 | 0 | 0 | 1 | 0 | 1 |
| Brown Hair | 0 | 0 | 0 | 1 | 0 | 0 | 0 | 1 | 0 | 1 |
| Black Hair | 0 | 1 | 0 | 0 | 0 | 0 | 0 | 1 | 0 | 1 |
| Male | 0 | 1 | 0 | 0 | 0 | 1 | 0 | 1 | 0 | 1 |
| Facial Hair | 0 | 1 | 0 | 0 | 0 | 1 | 1 | 1 | 0 | 1 |
| Eyeglasses | 0 | 1 | 0 | 0 | 1 | 1 | 1 | 1 | 0 | 1 |
| Bald | 1 | 0 | 0 | 0 | 0 | 1 | 1 | 1 | 0 | 1 |
| Aged | 1 | 0 | 0 | 0 | 0 | 1 | 1 | 0 | 0 | 0 |

Table 3: Complete list of attributes for label names in Figures 4, 7, and 8

.

| $\sigma_x^2$ | LL | KL | ELBO |
|---|---|---|---|
| 1 | -11360 | 30 | -11390 |
| 1e-1 | -11325 | 150 | -11475 |
| **1e-2** | 15680 | 600 | **15080** |
| 1e-3 | 16090 | 1950 | 14140 |
| 1e-4 | 16150 | 3650 | 12500 |

Table 4: Selection of $\sigma_x = 0.1$ for the CelebA VAEs by ELBO maximization. All results are given in Nats.

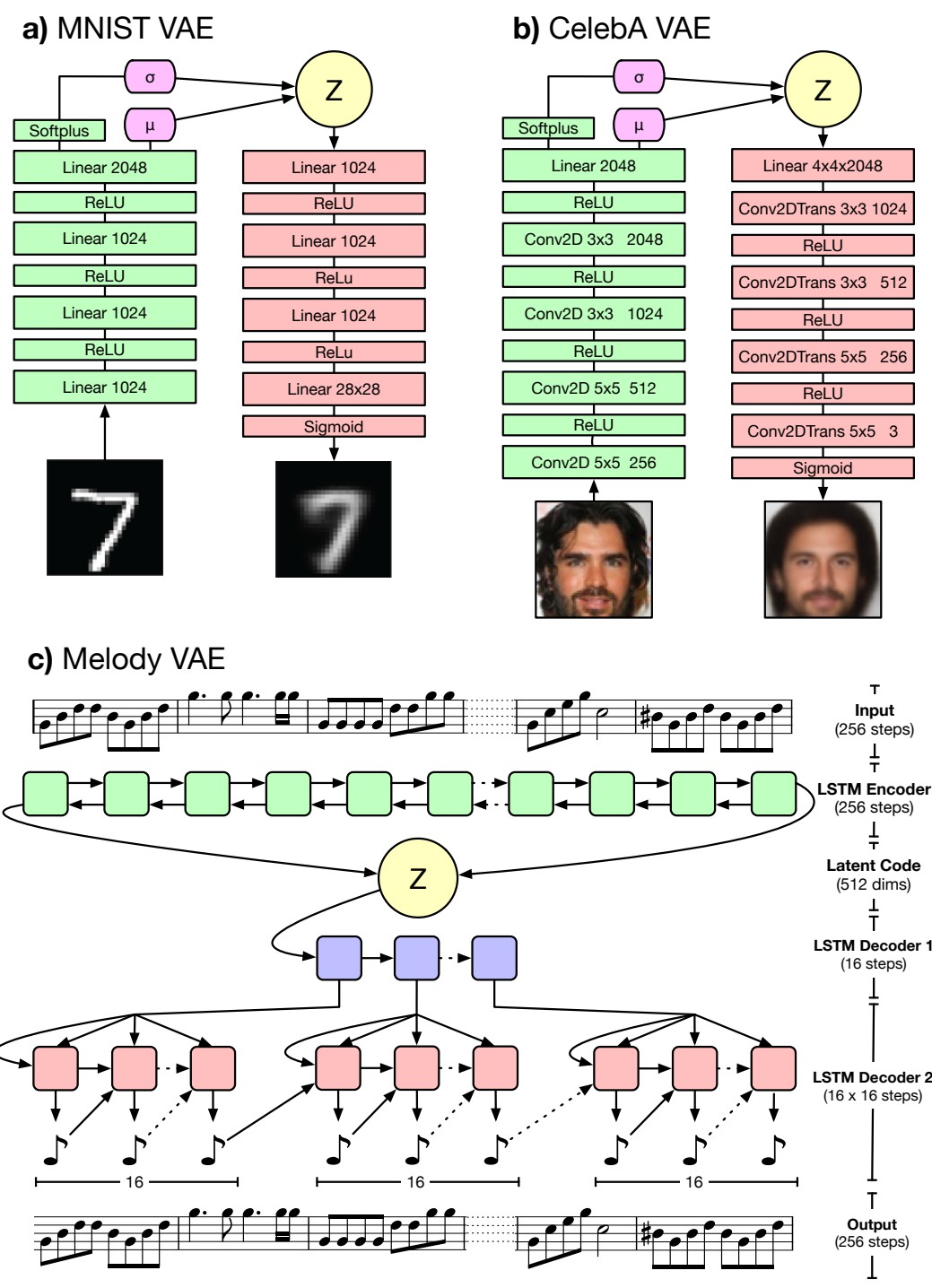

Figure 13: Architectures for the (a) feed-forward MNIST, (b) convolutional CelebA, and (c) hierarchical LSTM melody VAEs. In (b), all convolutions have a stride of 2. In (c), LSTM cells shown in the same color share weights and linear layers between levels are omitted.

## Small G/D Models (256 ReLU x 3)

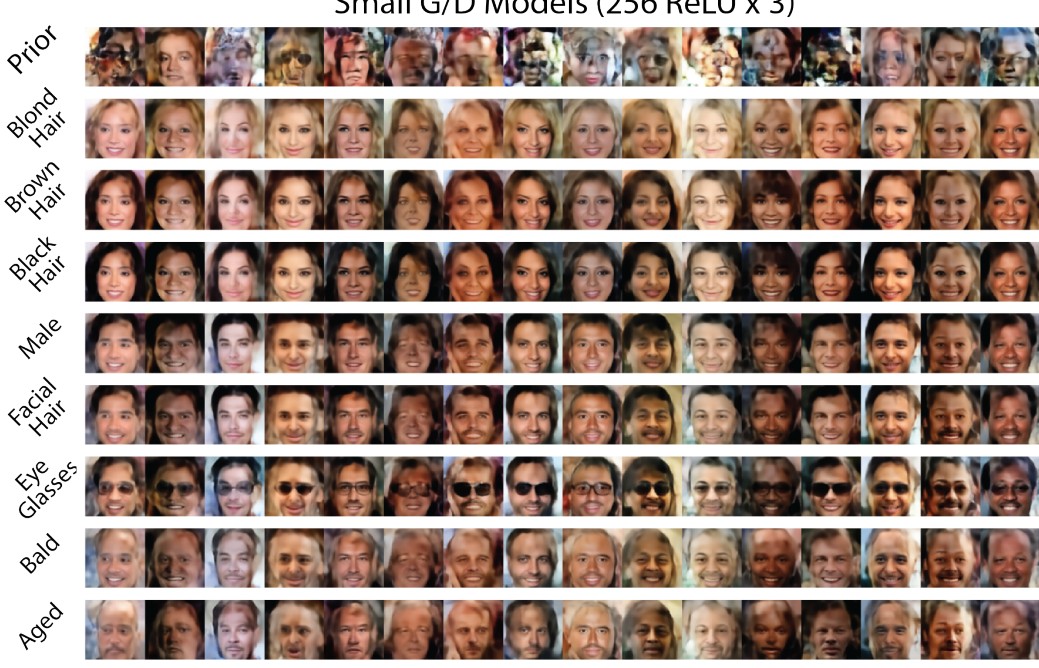

Figure 14: Samples generated with smaller (3 ReLU layers of 256 units each) $G$ and $D$ models are comparable quality despite having 85x fewer parameters, $\lambda_{\text{dist}} = 0.0$. Full attribute labels are given in supplementary Table 3.

## No KL Divergence Cost

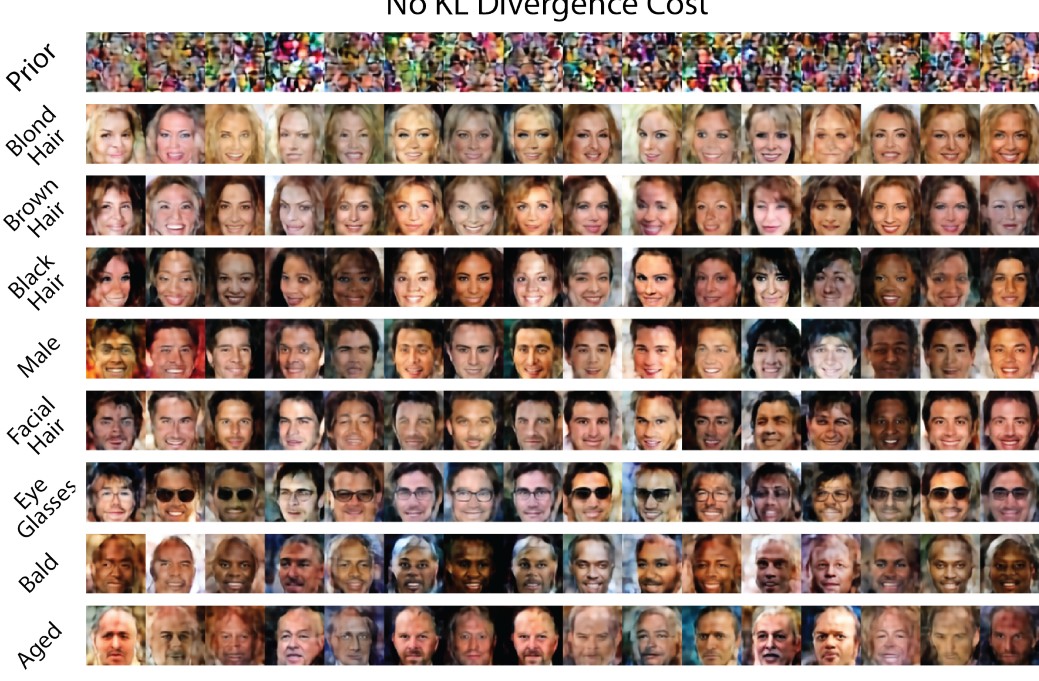

Figure 15: Latent constraints applied to a vanilla autoencoder with no latent prior. Samples are similar quality to VAEs with $\sigma_x = 0.1$, but with less diversity and more high-frequency visual artifacts. Full attribute labels are given in supplementary Table 3.

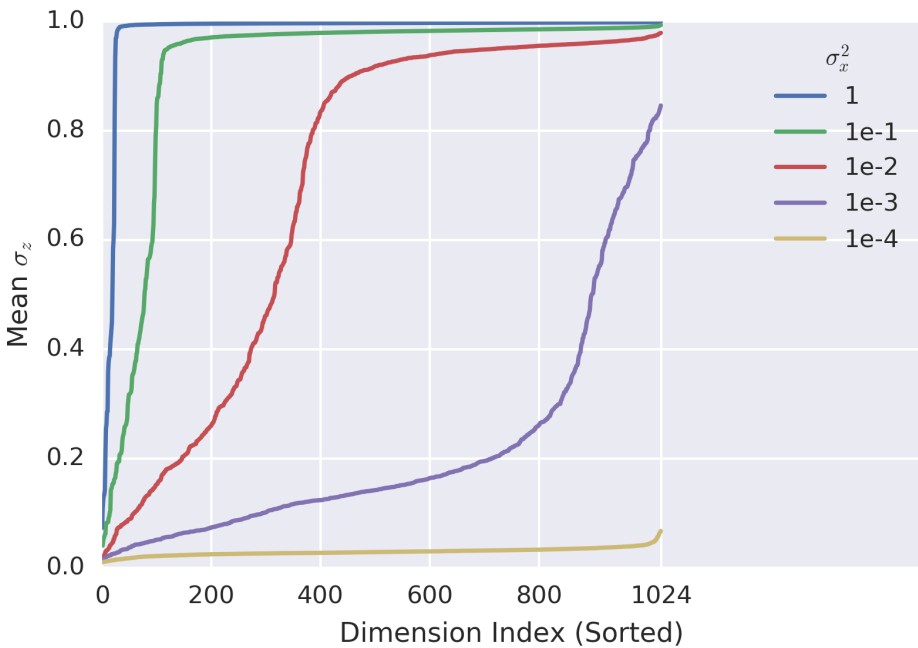

Figure 16: Smaller decoder standard deviations, $\sigma_x$, lead to lower-variance posteriors, $\sigma_z(x)$ of the encoder $q(z \mid x)$, averaged over the training set per a dimension. The x-axis is sorted from lowest to highest variance. Tighter posteriors correspond to more utilization of the latent dimension, and we scale our distance regularization the square inverse on a per-dimension basis.

