# OpenReview forum: "Latent Constraints: Learning to Generate Conditionally from Unconditional Generative Models"
_ICLR.cc/2018/Conference — Accept (Poster)_

### Official Review · AnonReviewer2 · 2017-11-27
**Seems like a solid paper on a timely topic**

**Rating:** 7
**Confidence:** 3

**Review:**

This paper considers the problem of generating conditional samples from unconditional models, such that one can query the learned model with a particular set of attributes to receive conditional samples.  Key to achieving this is the introduction of a realism constraint that encourages samples to be more realistic without degrading their reconstruction and a critic which identifies regions of the latent space with targeted attributes.   Generating conditional samples then involves finding points in the latent space which satisfy both the realism constraint and the critic.  This is carried out either used gradient-based optimization or using an actor function which tries to amortize this process.

This paper is clearly on a timely topic and addresses an important problem.  The low-level writing is good and the paper uses figures effectively to explain its points.  The qualitative results presented are compelling and the approaches taken seem reasonable.  On the downside, the quantitative evaluation of method does not seem very thorough and the approach seems quite heuristical at times. Overall though, the paper seems like a solid step in a good direction with some clearly novel ideas.

My two main criticisms are as follows
1. The evaluation of the method is generally subjective without clear use of baselines or demonstration of what would do in the absence of this work - it seems like it works, but I feel like I have a very poor grasp of relative gains.  There is little in the way of quantitative results and no indication of timing is given at any point.  Given that the much of the aim of the work is to avoid retraining, I think it is clear to show that the approach can be run sufficiently quickly to justify its approach over naive alternatives.

2. I found the paper rather hard to follow at times, even though the low-level writing is good.  I think a large part of this is my own unfamiliarity with the literature, but I also think that space has been prioritized to showing off the qualitative results at the expense of more careful description of the approach and the evaluation methods.  This is a hard trade-off to juggle but I feel that the balance is not quite right at the moment.  I think this is a paper where it would be reasonable to go over the soft page limit by a page or so to provide more precise descriptions.  Relatedly, I think the authors could do a better job of linking the different components of the paper together as they come across a little disjointed at the moment.

---

> ### Author Response · Authors · 2017-12-21
> **Response to AnonReviewer2**
>
> Thank you for your time and expertise in your review, we've addressed the key points below:
>
> > “Given that the much of the aim of the work is to avoid retraining, I think it is clear to show that the approach can be run sufficiently quickly to justify its approach over naive alternatives.”
>
> Thank you for highlighting that computational efficiency is indeed one of the strengths of this approach. Retraining the whole VAE can be shortcut by training a much smaller and less expensive actor-critic pair on user preferences. While our initial experiments did not focus on computational efficiency, we have since repeated the experiments and found similar results (Supplemental Figure 14 and Table 1), are achievable with a much smaller model (~85x fewer parameters than original generator / discriminator, ~2884x fewer FLOPS/iter than training the VAE). We have updated Table 1 and the main text to emphasize this.
>
>
> > 2. “...I think the authors could do a better job of linking the different components of the paper together as they come across a little disjointed at the moment”
>
> We agree that there are several interwoven elements to the story. To better summarize and clarify the experimental design we have overhauled and streamlined the visual depiction in Figure 1.

---

> > ### Comment · AnonReviewer2 · 2017-12-24
> > **Follow up**
> >
> > Thanks for your response and the revision.  I like the updates, particularly Figure 1 which I think is now much better.  Just in case it wasn't clear, I had a somewhat silly typo in my original review where it should have read "I think it is key" rather than "I think it is clear", sorry about that.  I'm happy with your updates to that comment anyway though.
> >
> > I still feel the paper is relatively weak in the level of quantitative comparison to previous methods.  However, I appreciate that such comparison is difficult to make for this work and I don't have any reasonable concrete suggestions to improve this.

---

### Official Review · AnonReviewer1 · 2017-11-27
**Nicely written paper: one key missing reference.**

**Rating:** 7
**Confidence:** 3

**Review:**

# Paper overview:
This paper presents an analysis of a basket of approaches which together enable one to sample conditionally from a class of
generative models which have been trained to match a joint distribution. Latent space constraints (framed as critics) are learned which confine the generating distribution to lie in a conditional subspace, which when combined with what is termed a 'realism' constraint enables the generation of realistic conditional images from a more-or-less standard VAE trained to match the joint data-distribution.

'Identity preserving' transformations are then introduced within the latent space, which allow the retrospective minimal modification of sample points such that they lie in the conditional set of interest (or not).  Finally, a brief foray into unsupervised techniques for learning these conditional constraints is made, a straightforward extension which I think clouds rather than enlightens the overall exposition.

# Paper discussion:
I think this is a nicely written paper, which gives a good explanation of the problem and their proposed innovations, however I am curious to see that the more recent "Plug & Play Generative Networks: Conditional Iterative Generation of Images in Latent Space" by Nguyen et al. was not cited.  This is an empirically very successful approach for conditional generation at 'test-time'.

Other minor criticisms include:
* I find the 'realism' constraint a bit weak, but perhaps it is simply a naming issue.  Did you experiment with alternative approaches for encouraging marginal probability mass?

* The regularisation term L_dist, why this and not log(1 + exp(z' - z)) (or many arbitrary others)?

* The claim of identity preservation is (to me) a strong one: it would truly be hard to minimise the trajectory distance wrt. the actual 'identity' of the subject.

* For Figure 6 I would prefer a different colourscheme: the red does not show up well on screen.

* "Furthermore, CGANs and CVAEs suffer from the same problems of mode-collapse and blurriness as their unconditional cousins" -> this is debateable, there are many papers which employ various methods to (attempt to) alleviate this issue.


# Conclusion:
I think this is a nice piece of work, if the authors can confirm why "Plug & Play Generative Networks: Conditional Iterative Generation of Images in Latent Space" is not placed relative to this work in the paper, I would be happy to see it published.  If stuck for space, I would personally recommend moving the one-shot generation section to the appendix as I do not think it adds a huge amount to the overall exposition.

---

> ### Author Response · Authors · 2017-12-21
> **Response to AnonReviewer1**
>
> Thank you for your review. We've incorporated changes to the paper and respond to your main points below:
>
>
> > “I am curious to see that the more recent "Plug & Play Generative Networks: Conditional Iterative Generation of Images in Latent Space" by Nguyen et al. was not cited.  This is an empirically very successful approach for conditional generation at 'test-time'.”
>
> * Thank you for highlighting the paper by Nguyen et al. It is indeed relevant and we have added a citation to the main text. For more context, we highlight several key differences between the papers here below.
> * PPGNs require a high-quality pretrained image-space classifier. This makes them less applicable to domains where very large labeled datasets are unavailable.
> * To generate samples, PPGNs need to apply iterative gradient-based optimization in image space, each step of which requires expensive backpropagation through both a powerful CNN classifier and the generator network. By contrast, our iterative optimization procedure is done entirely in the latent space, which allows our critic networks to be much smaller and dramatically reduces the cost per iteration. Furthermore, our amortized GAN-based sampling approach can generate samples with no iterative optimization at all.
> * PPGNs must backprop through the full generative process, which limits their ability to use non-differentiable generator networks such as the autoregressive VAE we discuss in section 6. Our approach easily handles this non-differentiability, because it operates entirely in the continuous latent space.
> * Finally, we feel that our approach is simpler than PPGNs, in which three DAEs are trained to minimize a stochastic four-term loss.
>
>
> > “I find the 'realism' constraint a bit weak, but perhaps it is simply a naming issue.  Did you experiment with alternative approaches for encouraging marginal probability mass?”
>
> We considered the name “marginal posterior constraint” which was more specific, but less concise. Since the marginal posterior, q(z), corresponds to real datapoints, we consider “realism” to be a fair name for an implicit constraint that makes samples more similar to q(z).
>
> As we note in the future work section, there are other ways to constrain sampling to the marginal posterior, such as learning an explicit autoregressive density model (indeed, van den Oord et al. proposed just such an approach in their very recent paper “Neural Discrete Representation Learning”, although they argued that using discrete latent variables was essential to their success). We feel that combining these approaches is an interesting avenue to consider, but for simplicity we focused on implicit constraints.
>
>
> > “The regularisation term L_dist, why this and not log(1 + exp(z' - z)) (or many arbitrary others)?”
>
> The form of L_dist was inspired by the log-density of a student-t distribution, which we chose because it penalizes outliers less than the more obvious MSE regularizer (we found MSE regularization to be quite sensitive to the hyperparameter lambda_dist). There are indeed a number of other similarly heavy-tailed functions that could work; we did not experiment extensively with these, but a better choice may well exist.
>
>
> > “The claim of identity preservation is (to me) a strong one: it would truly be hard to minimise the trajectory distance wrt. the actual 'identity' of the subject.”
>
> Indeed, without conditioning on explicit labels it is hard to rigorously define identity preservation, let alone enforce it. We use the phrase “identity preserving” to emphasize that we are trying to match not only attributes, but also whatever other latent structure was discovered by the unconditional generative model. Empirically, we feel this approach produces results that, while not perfect, match intuitive notions of identity preservation much better than models that only attempt to match attributes.
>
>
> > “For Figure 6 I would prefer a different colourscheme: the red does not show up well on screen.”
>
> Point well taken. We’ve changed to grey so that it looks like an extension of the black keys on the piano and contrasts more with the red notes which are out of the key of C.

---

### Official Review · AnonReviewer3 · 2017-11-28

**Rating:** 7
**Confidence:** 4

**Review:**

UPDATE: I think the authors' rebuttal and updated draft address my points sufficiently well for me to update my score and align myself with the other reviewers.

-----

ORIGINAL REVIEW: The paper proposes a method for learning post-hoc to condition a decoder-based generative model which was trained unconditionally. Starting from a VAE trained with an emphasis on good reconstructions (and at the expense of sample quality, via a small hard-coded standard deviation on the conditional p(x | z)), the authors propose to train two "critic" networks on the latent representation:

1. The "realism" critic receives either a sample z ~ q(z) (which is implicitly defined as the marginal of q(z | x) over all empirical samples) or a sample z ~ p(z) and must tell them apart.
2. The "attribute" critic receives either a (latent code, attribute) pair from the dataset or a synthetic (latent code, attribute) pair (obtained by passing both the attribute and a prior sample z ~ p(z) through a generator) and must tell them apart.

The goal is to find a latent code which satisfies both the realism and the attribute-exhibiting criteria, subject to a regularization penalty that encourages it to stay close to its starting point.

It seems to me that the proposed realism constraint hinges exclusively on the ability to implictly capture the marginal distribution q(z) via a trained discriminator. Because of that, any autoencoder could be used in conjunction with the realism constraint to obtain good-looking samples, including the identity encoder-decoder pair (in which case the problem reduces to generative adversarial training). I fail to see why this observation is VAE-specific. The authors do mention that the VAE semantics allow to provide some weak form of regularization on q(z) during training, but the way in which the choice of decoder standard deviation alters the shape of q(z) is not explained, and there is no justification for choosing one standard deviation value in particular.

With that in mind, the fact that the generator mapping prior samples to "realistic" latent codes works is expected: if the VAE is trained in a way that encourages it to focus almost exclusively on reconstruction, then its prior p(z) and its marginal q(z) have almost nothing to do with each other, and it is more convenient to view the proposed method as a two-step procedure in which an autoencoder is first trained, and an appropriate prior on latent codes is then learned. In other words, the generator represents the true prior by definition.

The paper is also rather sparse in terms of comparison with existing work. Table 1 does compare with Perarnau et al., but as the caption mentions, the two methods are not directly comparable due to differences in attribute labels.

Some additional comments:

- BiGAN [1] should be cited as concurrent work when citing (Dumoulin et al., 2016).
- [2] and [3] should be cited as concurrent work when citing (Ulyanov et al., 2016).

Overall, the relative lack of novelty and comparison with previous work make me hesitant to recommend the acceptance of this paper.

References:

[1] Donahue, J., Krähenbühl, P., and Darrell, T. (2017). Adversarial feature learning. In Proceedings of the International Conference on Learning Representations.
[2] Li, C., and Wand, M. (2016). Precomputed real-time texture synthesis with markovian generative adversarial networks. In European Conference on Computer Vision.
[3] Johnson, J., Alahi, A., and Fei-Fei, L. (2016). Perceptual losses for real-time style transfer and super-resolution. In European Conference on Computer Vision.

---

> ### Author Response · Authors · 2017-12-21
> **Response to AnonReviewer3**
>
> Thank you for your time and insight in your review. We've done our best to address your concerns with paper revisions and in the comments below:
>
>
> > ”the way in which the choice of decoder standard deviation alters the shape of q(z) is not explained, and there is no justification for choosing one standard deviation value in particular.”
>
> This was not made sufficiently clear in the original version. We chose a standard deviation parameter of 0.1 because it maximizes the ELBO. Using ELBO maximization as a hyperparameter selection scheme is a very natural and well-established practice (cf. Bishop's 2006 Pattern Recognition and Machine Learning textbook, for example). We have updated the text to highlight this and added Table 4 to the appendix, which shows the very significant improvement in ELBO from sigma=1.0 to sigma=0.1.
>
>
> > “any autoencoder could be used in conjunction with the realism constraint to obtain good-looking samples…”
>
> This is true enough, and worth emphasizing—our contributions are not specific to VAEs, but can be used to generate good-looking conditional samples from pretrained classical autoencoders. We have added Figure 15 to the supplement, which explores what happens when the VAE’s sigma parameter goes to 0 (equivalent to a classical autoencoder). We obtain reasonably good conditional samples with high-frequency spatial artifacts.
>
> We focused on VAEs rather than classical AEs both because they have natural sampling semantics and because they produced slightly better results. We believe this is because the KL divergence term encourages q(z) to fill up as much of the latent space as possible (without sacrificing reconstruction quality). This penalty encourages more of the latent space to map to reasonable-looking images.
>
>
> > “...including the identity encoder-decoder pair (in which case the problem reduces to generative adversarial training).”
>
> This is an interesting observation, and may be true for the simplest version of our approach (although the identity mapping would stretch the definition of “latent” space). But it breaks down when we regularize the GAN to not move too far from the input z vector, which we found was essential to combat mode collapse and find identity-preserving transformations. In that case, it is essential that Euclidean distance in latent space be more meaningful than distance in pixel space, making the identity “autoencoder” a poor choice.
>
>
> > “the way in which the choice of decoder standard deviation alters the shape of q(z) is not explained”
>
> Smaller standard deviations will lead to lower-variance posteriors, and therefore a more concentrated q(z). This may not be obvious to all readers, so we updated the text to emphasize it, and added Supplemental Figure 16, which demonstrates the effect experimentally.
>
>
> > “The paper is also rather sparse in terms of comparison with existing work. Table 1 does compare with Perarnau et al., but as the caption mentions, the two methods are not directly comparable due to differences in attribute labels.”
>
> We do our best to find work with which to compare, and match experimental conditions, however, there are not well established benchmarks for this type of task. Unfortunately, Perarnau et al. do not list the specific attributes that they selected as most salient, so an exact comparison is not possible. We do our best to match conditions, and provide a list our 10 salient features in supplemental Table 3 for future comparison.
>
>
> > “- BiGAN [1] should be cited as concurrent work when citing (Dumoulin et al., 2016).  [2] and [3] should be cited as concurrent work when citing (Ulyanov et al., 2016).”
>
> Thank you for bringing these citations to our attention. They are indeed concurrent work with Dumoulin et al.’s and Ulyanov et al.’s work, and we have cited them as such.

---

### Decision · Program_Chairs · 2018-01-29
**ICLR 2018 Conference Acceptance Decision**

**Decision:**

Accept (Poster)

**Comment:**

This paper clearly surveys a set of methods related to using generative models to produce samples with desired characteristics.  It explores several approaches and extensions to the standard recipe to try to address some weaknesses.  It also demonstrates a wide variety of tasks.  The exposition and figures are well-done.